# Enhancing the spatio-temporal features of polar mesosphere summer echoes using coherent MIMO and radar imaging at MAARSY

Juan Miguel Urco<sup>1</sup>, Jorge Luis Chau<sup>1</sup>, Tobias Weber<sup>2</sup>, and Ralph Lateck<sup>1</sup>

<sup>1</sup>Leibniz Institute of Atmospheric Physics at the University of Rostock, Germany <sup>2</sup>University of Rostock, Germany

Correspondence: J.M. Urco (urco@iap-kborn.de)

**Abstract.** Polar mesospheric summer echoes (PMSEs) are very strong radar echoes caused by the presence of ice particles, turbulence, and free electrons in the mesosphere over polar regions. For more than three decades, PMSEs have been used as natural tracers of the complicated atmospheric dynamics of this region. Neutral winds and turbulence parameters have been obtained assuming PMSE horizontal homogeneity in scales of tens of kilometers. Recent radar imaging studies have shown

- that PMSEs are not homogeneous in these scales and instead they are composed of kilometer-scale structures. In this paper, we present a technique that allows PMSE observations with unprecedented angular resolution ( $\sim 0.6^{\circ}$ ). The technique combines the concept of coherent MIMO (Multi-input multiple-output) and two high-resolution imaging techniques, i.e., Capon and Maximum Entropy (MaxEnt). The resulting resolution is evaluated by imaging specular meteor echoes. The gain in angular resolution compared to previous approaches using SIMO (single-input and multiple-output) and Capon is at least a factor of
- 2, i.e., at 85 km, we obtain a horizontal resolution of ~ 900 meters. The advantage of the new technique is evaluated with two events of three-dimensional PMSE structures showing: (1) horizontal wavelengths of 8-10 km and periods of 4-7 minutes, drifting with the background wind, and (2) horizontal wavelengths of 12-16 km and periods of 15-20 minutes not drifting with the background wind. Besides the advantages of the implemented technique, we discuss its current challenges, like the use of reduced power-aperture and processing time, as well as the future opportunities for improving the understanding of the
- complex small-scale atmospheric dynamics behind PMSEs.

Copyright statement. TEXT

#### 1 Introduction

The so called MIMO (Multiple Input Multiple Output) technique is being widely used in the fields of telecommunications and radar remote sensing (e.g Telatar, 1999; Huang et al., 2011; Foschini and Gans, 1998). Recently Urco et al. (2018) have shown that the use of multiple-transmitters and multiple-receivers can significantly improve the angular resolution of coherent atmospheric/ionospheric radars. In that work, MIMO was used to observe equatorial electrojet (EEJ) field-aligned irregularities at Jicamarca in combination with the well-established radar imaging technique Capon (e.g., Palmer et al., 1998). The multiple transmitter part was implemented with three different diversity schemes, i.e., temporal, code, and polarization. The resulting angular resolution was superior, at least a factor of 4 to previous efforts using a single transmitter and the same receiving configuration, i.e., SIMO. Given that the EEJ irregularities are field-aligned with the Earth's magnetic field, angular imaging uses performed only in the magnetically. East West direction

was performed only in the magnetically East-West direction.

Based on this successful implementation, we decided to implement coherent MIMO to improve the angular resolution of the Middle Atmosphere ALOMAR Radar System (MAARSY) (16.04°E, 69.30°N) and to study polar mesospheric summer echoes (PMSEs). PMSEs present strong radar cross sections (RCSs) that allow observing them with less transmitting power, which is the case when using MIMO. Previous efforts to study their spatial structure have been limited to a few kilometers spatial and

10 a few minutes temporal resolutions (e.g., Yu et al., 2001; Latteck et al., 2012a; Stober et al., 2013; Sommer and Chau, 2016). Recently, Stober et al. (2018) has presented many examples of monochromatic gravity waves (GWs) and Kelvin-Helmholtz instabilities (KHIs) using nine days of multi-beam PMSE observations with MAARSY.

PMSEs are strong echoes, more than 50 dB stronger than expected echoes from free electrons in the D region, and there is a consensus that they are generated by atmospheric turbulence and require the presence of free electrons and charged ice

- particles (e.g., Rapp et al., 2002; Varney et al., 2011, and references therein). Although PMSEs have been studied since the late 1970s (e.g., Ecklund and Balsley, 1981; Hoppe et al., 1988; Kelley and Ulwick, 1988; Havnes et al., 1996; Rapp and Lübken, 2004), until recently they have been considered very aspect sensitive and homogeneous in scales of a few tens of kilometers at least when observed at very high frequencies (VHF) (e.g. Czechowsky et al., 1988; Zecha et al., 2001; Yu et al., 2001).
- Based on recent multi-beam observations as well as radar imaging, Sommer and Chau (2016) have concluded that the PMSEs
  are not as aspect sensitive as previously reported, and instead, they are most of the time organized in kilometer-scale spatial structures drifting across the observing beams. Such results have been independently verified with bistatic observations at VHF, where PMSEs were observed with small systems at zenith angles close to 30° (e.g., Chau et al., 2018).

The results of Sommer and Chau (2016) were obtained with MAARSY using the whole antenna array for transmitting and an antenna compression approach, i.e., a wide beam by properly phasing the antennas (e.g., Woodman and Chau, 2001), and

a multiple-receiver configuration. The spatial structures were obtained with the Capon technique due to its implementation simplicity and its relatively fast processing speed.

Given that PMSE, are highly associated to Noctilucent clouds (NLCs) (e.g., Hoppe et al., 1990; Stebel et al., 2000; Kaifler et al., 2011), spatial structures ranging from a few hundreds of meters to a few tens of kilometers observed in NLCs (e.g., Baumgarten and Fritts, 2014) are expected to be observed also in PMSEs. Indeed this is the case, PMSE structures of a few

kilometers have been already reported by Sommer and Chau (2016) and structures of a few tens of kilometers have been reported by Chau et al. (2018).

Although progress has been made in discriminating between spatial and temporal ambiguities in PMSE observations, the achieved angular resolution has been mainly limited by two factors: (1) the effective area in the visibility plane, and (2) the number of independent spatial samples (e.g., Woodman, 1997). By implementing MIMO, we are able to improve both, i.e., a

35 larger effective area, and a higher number of independent visibility samples. In addition, by implementing Maximum Entropy

(MaxEnt) which is more computationally demanding than Capon, we are able to further improve the angular resolution (e.g., Hysell and Chau, 2006).

In this work, we have implemented coherent MIMO at MAARSY using three spatially separated antenna sections on transmission and fifteen on reception. Moreover, time diversity was employed in order to isolate radar echoes corresponding to each

5 transmitting section, i.e., the transmitters were interleaved every 4ms. The resulting effective number of virtual receivers by using MIMO was 45 and the angular resolution achieved was  $\sim 0.6^{\circ}$ . It is equivalent to an antenna area of 450 m diameter, more than 5 times larger than the nominal diameter of the MAARSY antenna.

Our paper is organized as follows. We first present the experiment configuration with a specific emphasis on the MIMO implementation. Then we describe the radar imaging implementation for both Capon and MaxEnt techniques. The PMSE

results are shown in Section 4 for SIMO and MIMO using both Capon and MaxEnt. Within this section, two events are studied in detail, one where the observed waves drift with the background wind, and a second one where the waves do not propagate with the wind. Finally, the results of our MIMO implementation are discussed followed by conclusions.

### 2 Experiment configuration

#### 2.1 MAARSY

- MAARSY is an active phased antenna array operating at 53 MHz located in Andoya, Norway  $(69.30^{\circ}N, 16.04^{\circ}E)$ . The array consists of 433 antenna elements, each with its own transceiver module that allow us to modulate the antennas in phase and amplitude independently. Using this capability the transmitting or receiving beam can be steered in a desired direction up to 30° off zenith with an angular resolution of  $3.6^{\circ}$  (e.g., Latteck et al., 2012a). In addition to its multibeam capability, MAARSY can be used for in-beam imaging experiments. In this case, the signals from a selected number of receiving antennas are
- stored and later a digital beamforming algorithm (imaging) is applied to the data. Unlike the multi-beam experiment, imaging allows obtaining a 2D image at once, avoiding the interleave from beam to beam. Currently, only 16 receivers are available at MAARSY. These 16 receive signals can be selected from groups of 7 antennas each called "hexagons" or from a group of 7 hexagons called "anemones" (e.g., Latteck et al., 2012b, for further technical details). For this campaign, we conducted an imaging experiment using 15 hexagons on reception similar to Sommer and Chau (2016)'s experiment. One receiver is
- always connected to the full antenna array and it is used as in the standard multiple experiments. The radar parameters of this experiment are summarized in Table 1.

#### 2.2 MAARSY MIMO configuration

In order to improve the performance of our imaging experiment, we applied a coherent MIMO technique [Urco et al. (2018)]. The technique employs multiple independent transmitting antennas and multiple receiving antennas, both spatially separated, to take advantage of the transmit-receive geometry and to increase the angular resolution of the radar. If the antennas are closely separated or collocated, the signals from each transmitting-receiving path are coherent and can be combined to form a larger virtual receiving array. The resulting number of virtual receivers is equal to the number of transmitters times the number of receivers.

Depending on the transmitting and receiving antenna configuration some virtual receivers can be redundant. In our experiment, we carefully selected the transmitting and receiving antenna configuration to get 3 special redundant virtual receivers.

These 3 redundant virtual receivers were used for phase calibration of the transmitters as it was done by Urco et al. (2018). 5 Figure 1a shows the 15 hexagons used in reception and the 3 anemones used in transmission (B, D, F). Figure 1d shows the resulting virtual receiving antennas where 3 of them are redundant and located at the origin.

In order to separate the contribution of each transmitter a form of transmit diversity was needed. In Urco et al. (2018). 3 types of transmit diversity were proposed: Code, time and polarization. Code diversity is recommended for atmospheric

- observations given that this is not sensitive to the temporal correlation or polarization of the target of interest. Unfortunately, 10 code diversity cannot be currently used in MAARSY. For targets where the temporal correlation is less than the time separation between transmitters, time diversity can be applied. Given that PMSEs have a relative long correlation time (a few hundreds of milliseconds) we applied time diversity to enhance the spatio-temporal features of PMSE. The effective time separation between transmitters was 4, 4, 8 milliseconds between pairs BD, DF, and BF, respectively.

As explained by (Urco et al., 2018), in a monostatic coherent MIMO radar the relationship between the normalized spatial cross-correlation of signals from two different transmitting-receiving paths and the angular distribution of scattered power for a given range and frequency bin can be described by:

$$\frac{\langle v_{mp} * v_{nq}^* \rangle}{\sqrt{\langle |v_{mp}|^2 \rangle \langle |v_{nq}|^2 \rangle}} = V \big( \boldsymbol{k} (\Delta \boldsymbol{r}_{mn} + \Delta \boldsymbol{r}_{pq}) \big)$$
(1)

$$=e^{(j2\pi f_d\tau + \phi_{mn} + \phi_{pq})} \cdot \int B(\theta) \cdot e^{-j\mathbf{k}(\Delta \mathbf{r}_{mn} + \Delta \mathbf{r}_{pq})} d\theta,$$
<sup>(2)</sup>

where:  $v_{mp}$  is the signal from the transmitting-receiving path (m,p), being m the receiver and p the transmitter;  $v_{nq}^*$  is 20 the complex conjugate of the signal from the transmitting-receiving path (n,q), being n the receiver and q the transmitter;  $\langle v_{mp} * v_{na}^* \rangle$  is the cross-correlation of two signals from antennas spatially separated;  $V(\mathbf{k}(\Delta \mathbf{r}_{mn} + \Delta \mathbf{r}_{pq}))$  is the visibility sample at  $(\Delta r_{mn} + \Delta r_{pq})$ ; k is the wave number vector equal to  $(2\pi/\lambda)\theta$ . And  $\lambda$  is the radar wavelength;  $\theta$  is the angle of arrival equal to  $(\theta_x, \theta_y, \theta_z)$  which are the direction cosines in the (x,y,z) direction;  $B(\theta)$  is the angular scattered power distribution, also known as brightness;  $\Delta r_{mn}$  is the spatial separation between receivers m and n;  $\Delta r_{pq}$  is the spatial separation 25 between transmitters p and q;  $2\pi f_d \tau$  is the phase difference due to the Doppler shift of the target,  $f_d$ , and  $\tau$  is the time separation between transmitters;  $\phi_{pq}$  is the phase difference between transmitters;  $\phi_{mn}$  is the phase difference between receivers.

A quick comparison between the visibility (sampling domain) for SIMO and MIMO shown in Figs. 1b and 1e, indicates that the antenna aperture for MIMO is larger than the SIMO by  $\sim 50\%$ . The difference lies in that the MIMO antenna aperture is defined as the maximum separation between two virtual receiving antennas, i. e.,  $Max(\Delta r_{mn} + \Delta r_{pq})$ . Whereas for SIMO 30  $\Delta r_{pq} = 0$  and the antenna aperture is only defined by the maximum spatial separation between two receiving antennas. Figures 1c and 1f show the resulting instrument function or point spread function (PSF) for SIMO and MIMO, respectively. As expected the half-power beam-width (HPBW) for MIMO is  $\sim 50\%$  smaller than for SIMO resulting in an angular resolution of 2.4° for

MIMO compared to 3.6° for SIMO. Furthermore, the sidelobes in the MIMO configuration are strongly reduced given that the visibility is larger and contains no gaps.

#### **3** Radar imaging implementation

Before inverting Eq. 2, the three phase differences due to time diversity  $(2\pi f_d \tau)$ , to receivers  $(\phi_{mn})$ , and to transmitters  $(\phi_{pq})$ need to be corrected. When the analysis is done in frequency domain we can easily correct the value  $2\pi f_d \tau$  given that we know the frequency and the time separation between transmitters. On the other hand, the phase offsets between receivers have been calibrated using Cassiopeia A as a radio source (e.g., Chau et al., 2014). Additionally, we have calibrated the phase offset between transmitters using the 3 redundant virtual receivers described above. Each of the redundant virtual receivers comes from one transmitter. They were compared to have zero phase difference between each other, given that they three must be located at the same virtual position (see, e.g., Urco et al., 2018, for more details).

Once the imaging system is calibrated we can invert Eq. 2 to obtain the estimated brightness  $B(\theta)$ . Given that the number M of unique visibility samples is still less than the number N of unknowns (Brightness points) some kind of regularization is needed to solve Eq. 2. Two of the most well known radar imaging techniques applied to atmospheric/ionospheric targets are Capon (Palmer et al., 1998) and MaxEnt (Hysell, 1996).

#### 15 3.1 Capon technique

As described by Kudeki and Sürücü (1991), the angular resolution obtained from a direct inversion of Eq. 2 using the inverse Fourier transform is limited by the longest baseline and the unmeasured antenna separations (visibility gaps). Palmer et al. (1998) proposed a new technique to improve the angular resolution based on the work of Capon (1969). Capon can be seen as an extension of the Fourier inverse transform. The difference lies in the fact that Capon chooses the antenna weights adaptively

in order to minimize the sidelobe interference from signals outside of the direction of interest according to the data. Capon's technique provides an estimate of the brightness function given by:

$$B(\hat{\theta}) = \frac{1}{M^H \cdot V^{-1} \cdot M},\tag{3}$$

where  $M = [e^{-jk(r_{m_0}+r_{p_0})}, e^{-jk(r_{m_0}+r_{p_1})}, \dots, e^{-jk(r_{m_1}+r_{p_0})}, \dots]^T$  is the Fourier kernel and  $V = V\{k(\Delta r_{m_im_j} + \Delta r_{p_kp_l}\}$  is the visibility due to the virtual receivers  $v_{m_ip_k}$  and  $v_{m_jp_l}$ , with i and j being the receiver indices and l and k being the transmitter indices.

## 3.2 Maximum Entropy technique

Even when MIMO is used, the problem is still underdetermined. Thus, there are infinite possible image solutions, B, which agree with the data, V. Of all possibilities, MaxEnt chooses the solution with the maximum entropy or minimal information

content (e.g., Hysell, 1996), as the one to be the most likely brightness distribution and the most consistent with the available visibility data and their statistical uncertainties. The entropy for a given frequency bin and range can be defined as:

$$S = \sum_{\theta} \hat{B(\theta)} \ln \{\hat{B(\theta)}/F\}$$

$$F = \sum_{\theta} \hat{B(\theta)}$$
(4)
(5)

5

where F is the summation of the brightness distribution over the region of interest. The solution of Eq. 2 is defined by:

$$\max_{\theta} \{S\} \quad \text{subject to} \quad |V - M \cdot \hat{B(\theta)}| 

The preliminary results using MIMO-MaxEnt are allowing us to observe PMSE with unprecedented horizontal resolution 25 (less than 1 km) compared to multibeam scanning experiments (Stober et al., 2013), and therefore the identification of structures with horizontal wavelengths less than 10 km (e.g., Event 1 above). For structures with wavelengths of the order of 15-20 km or so, the other imaging implementations, i.e., SIMO-Capon, SIMO-MaxEnt, MIMO-Capon, are sufficiently good to characterize them. These new capabilities will allow to better identify and characterize KHIs and general GWs (not only monochromatic) than previously done at polar mesospheric heights during the summer. Our proposed technique complements previous obser-

30

vations that have been performed at nighttime when the sky is clear using airglows and lidars (e.g., Smith, 2013; Hecht et al., 2000, 2007; Taylor et al., 2007).

We will leave the detailed analysis and interpretation of these events and other events observed with this new capability for a future effort. In the following paragraphs, we discuss the technical results and propose future improvements.

The improved resolution using MIMO results from the larger effective visibility aperture and the larger number of independent samples, as compared to a SIMO configuration, i.e., 125 m instead of 76 m, and 475 samples instead of 163 samples. respectively. In addition, the MaxEnt approach allows an improvement at least a factor of two in angular resolution when is compared to Capon. The maximum number of horizontal blobs that could theoretically be estimated for each range, time, and

- 5 "color" (i.e., frequency bin) would be 79 (=475/6), where each blob is characterized by a two-dimensional Gaussian function with 6 parameters (e.g., Chau and Woodman, 2001). Another reason for the better results using MIMO-MaxEnt is the number of redundant visibility measurements. Although they do not provide additional information in terms of degrees of freedom, the redundancy helps to reduce the statistical uncertainties of such visibility samples. Recall in our MIMO implementation, there are 1980 visibility samples (45x44) and only 475 are independent.
- Despite the significant improvement, not everything is positive about applying MIMO. In the following paragraphs, we 10 discussed the critical points of applying MIMO in terms of (a) power-aperture reduction, and (b) computational demands and real-time applicability.

As indicated by Urco et al. (2018), in atmospheric radars MIMO is applicable to targets with a large RCS, since a reduction of power-aperture is inherent to MIMO. In our particular application to PMSE, the transmitter sections were 1/7th of the total

- area, and therefore also 1/7th of the total transmitter power, i.e., -17 dB transmitting signal than usual experiments. In reception, 15 15 groups of 7 antennas (Hexagons) were used instead of the 433 available antennas. Moreover, given the time multiplexing, the number of coherent integrations was reduced and therefore the noise was increased, when compared to standard operations. In total, the sensitivity of our MIMO experiment is 27 dB less. Looking at the PMSE RCS in figure 2 of Latteck and Strelnikova (2015), our MIMO observations are limited to PMSE with RCS larger than  $10^{-14}$  m<sup>-1</sup>, i.e., approximately 40% of the usual
- seasonal MAARSY PMSE observations. 20

MaxEnt is known to be computationally more demanding than Capon in SIMO applications (e.g., Yu et al., 2000). In the case of MIMO, the computational demands are significantly increased given the larger number of effective receivers, i. e., 45 instead of 15. In terms of visibility pairs, the increase is from 210 to 1980! In the case of Capon, real-time processing is still possible with these increased numbers of samples, however, MaxEnt for both SIMO and MIMO is not applicable in a real-time

- 25 application. For example, for 80 seconds of data using an i5 PC with 15 cores, the processing times are 20 min and 3 hours for SIMO-MaxEnt and MIMO-MaxEnt, respectively. A future improvement to make MIMO-MaxEnt faster would be to use only one value of each redundant visibility sample, i.e., to work only with 475 independent samples instead of all 1980 measured visibility samples. Such value could be obtained either from the average of all the values sampling the same visibility or preselecting only one of them. After all, many of the independent samples are obtained with only one sample (green dots in Fig.
- 30 1e).

In general, a critical point for PMSE imaging is the drifting nature of the echoes. PMSE correlation times are relatively short, and under stationary conditions, one would require a few minutes of incoherent integration to reduce the statistical uncertainties of the visibility estimates. However, the structures to image would move between 2 and 5 km in 60 seconds for typical mesospheric motions (40-80 m/s), either from drifting with the background wind (Event 1) or from wave propagation

(Event 2). These drifting structures limit further the angular resolution that can be accomplished by any method since the resulting image will be significantly blurred for integration times of a few minutes.

To deal with the drifting nature of PMSEs, in future studies we will explore tracking techniques, i.e., make use of this information to improve the angular resolution (e.g., Vaswani and Zhan, 2016). Given the computational demands of MaxEnt 5 in particular when combined with MIMO, we will also explore radar imaging with compressed sensing (CS) techniques (e.g., Donoho, 2006; Candes and Wakin, 2008). Harding and Milla (2013) applied CS to Jicamarca F region irregularities, and show that CS produces results similar to MaxEnt. Our plan is to use MIMO-MaxEnt as a reference to other radar imaging

techniques using SIMO, for example, CS in combination with tracking. Besides the computational demands, MIMO might not be applicable at other atmospheric radar sites, and therefore the exploration of other techniques using SIMO is required.

An additional improvement to the current observations would be the use of shorter pulses and therefore better range resolution, for example, 150 meters. Further improvement in range could be accomplished also by applying range imaging (e.g., Palmer et al., 1998; Yu and Palmer, 2001), particularly in combination with the radar imaging implementations of this work, allowing angular resolutions less than 1°.

#### 6 Conclusions

- In this work, we have successfully implemented coherent MIMO with radar imaging at MAARSY to observe PMSEs with unprecedented angular resolution. The obtained resolution results from the combination of a larger effective aperture, higher number of independent visibility samples resulting from MIMO, and improved angular resolution resulting from MaxEnt. Quantitatively, the maximum angular resolution accomplished is  $\sim 0.6^{\circ}$ , which is equivalent to having a 450-m diameter visibility aperture at 53.5 MHz and significant improvement to the MAARSY standard angular resolution of  $3.6^{\circ}$ .
- The preliminary results with MIMO-MaxEnt allowed to clearly identify structures slightly less than 1 kilometer in diameter and wave-like structures with horizontal wavelengths less than 10 km, with a time resolution around 60 seconds. The identification of such structures with varying degrees of intensity, suggests that one has to be careful about using PMSE for estimating the background wind assuming horizontal homogeneity. Not only the vertical wind is not homogeneous, but also the brightness is not homogeneous horizontally.
- Given the relatively long temporal correlation of PMSEs, i.e., a few minutes, larger integration in time of the noisy visibility would allow less statistical uncertainties in the resulting images of the two events presented. However, PMSE structures drift as they are imaged, therefore long integration times result in angular smearing. In the future, we plan to use the drifting information to improve the angular resolution, by applying tracking techniques.

As mentioned above, the implementation of MIMO-MaxEnt is computationally intensive and is currently not applicable to real-time processing. On the other hand, MIMO-Capon can be implemented in real-time processing. Our strategy for near future observations would be to use MIMO-Capon for real-time processing and on special events use MIMO-MaxEnt until more efficient implementations and/or faster computers are available. *Acknowledgements.* We would like to thank Toralf Renkwitz for providing the receivers' phase offsets and Marius Zecha for MAARSY data handling. This work was partially supported by the Deutsche Forschunggemeinschaft (DFG, German Research Foundation) under SPP 1788 (CoSIP)-CH1482/3-1 and by the WATILA Project (SAW-2015-IAP-1).

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
