# Peer review of "Enhancing the spatio-temporal features of polar mesosphere summer echoes using coherent MIMO and radar imaging at MAARSY"

_Atmospheric Measurement Techniques, 2018_

## Referee Comment (RC1) · J. Yue (Referee) · 28 Aug 2018

This paper introduces the new technique of MIMO and two radar imaging methods to improve the spatial and temporal resolution of PMSE imaging. Two PMSE events are presented. One shows small features drifting with winds, and another displays gravity waves. The paper is well written. I could recommend its publication after minor revisions.

1. By inspecting Figures 3-6, the improvement from SIMO to MIMO is significantly smaller than that between Capon and MaxEnt. Can the authors comment on it? 2. The small features drifting with winds in event 1 seem like so called ripples from gravity

wave breaking. Ripples often show up in airglow and PMCs.

Hecht (2004), Instability layers and airglow imaging, Review of Geophysics.

- 3. page 1, line 10 "goodness" -> "advantage"
- 4. page 3, line 16, "e.g., see Latteck..."
- 5. page 10, line 9, "e.g., M. 2013"?
- 6. page 10, line 14, "475 m instead of 163 m".
- 7. The fond sizes in Figure 4 and 6 need to be enlarged.

Jia Yue

---

## Short Comment (SC1) · 29 Aug 2018

We would like to thank the reviewer for his useful and positive comments. We will address them in detail in the final version. Here we would like to comment on a couple of them:

1. "By inspecting Figures 3-6, the improvement from SIMO to MIMO is significantly smaller than that between Capon and MaxEnt. Can the authors comment on it?"

The expected improvement from SIMO to MIMO is about 50% due to the resulting antenna configuration by using MIMO. See Fig. 1. It could be further improved by

using a longer separation between the transmitting antennas but we are limited to the currently installed antenna array. This improvement can be quantitatively verified in Table 2 for a point-like target.

As described in P.7 line 1, the Capon performance is strongly affected by the increase in the number of targets. For the two events showed in this paper the illuminated area is full-filled with PMSE, so a poor performance is expected when Capon is used. However, we still expect an improvement of ~50% from SIMO-Capon to MIMO-Capon. In the case of MaxEnt, the algorithm is limited by the amount of information contained in the image (Entropy), i. e., changes in the image.

2. "The small features drifting with winds in event 1 seem like so-called ripples from gravity wave breaking. Ripples often show up in airglow and PMCs."

We agree that the description of ripples fit with our observations (Event 1). We will modify our text accordingly, including the suggested reference.

---

## Referee Comment (RC2) · I.W. McCrea (Referee) · 18 Oct 2018

This is an interesting paper which shows convincingly how the effective resolution of radar targets with large backscatter cross-section, such as PMSE layers, can be significantly improved by modularising the radar aperture and using MIMO (multi-input, multi-output techniques) combined with high-resolution image processing strategies, such as Capon and Maximum Entropy. The resulting improvements in image resolution are quite striking, even for a relatively small modular radar aperture such as MAARSY, and clearly open up the possibility of understanding the structure and dynamics of PMSE layers at higher temporal and spatial resolution, with the potential to

add to the current understanding of the physics of these layers. The techniques themselves are well described and in the case of the image processing strategies, have been extensively documented in previous literature. The implementation is seemingly based on earlier experiments at Jicamarca, which have attracted quite a lot of interest in the community.

The results clearly show intensity variations in the PMSE layers corresponding to wavelike activity, which are plausibly linked to generation by the Kelvin-Helmholtz Instability and display wind-related dynamics. This dynamics is, however, somewhat puzzling, because in one example the waves seem to drift with the background wind, while in a second case they do not. I found the discussion about the relationship between the phase front orientation, the drifting of the wave field and the strength and direction of the background wind somewhat confusing, because I was unsure exactly how to interpret Figure 7. The text seems to indicate that the "arrow slope" indicates the magnitude of the wind velocity, when this is normally the arrow length. Hence I am unsure how to interpret the arrow length and direction in terms of vector velocity. For example in event 1, the wind is apparently northward, but in Figure 7(b) the zonal wind vectors also appear substantial (at least the arrows are long in Fig 7b, which shows the zonal component). Also, in Figures 7(b) and 7(c), there appears to be wave front structure in both the meridional and zonal directions, whereas one might expect a KHI wave field driven by a meridional wind to have zonally-oriented phase fronts. I think this figure needs a clearer explanation to make it more intuitive to the reader. Nonetheless the results are clearly very interesting and seem to offer significant potential for a more physics-based study.

The MIMO technique combined with Maximum Entropy imaging clearly shows smaller structures than the SIMO techniques, or even MIMO plus Capon (such small structure are notable in Figures 3d, 8c and 9c), the authors should perhaps say something about their persistence and statistical significance. The text implies that some of them may be meteor echoes, but this point is not discussed in detail. As a reviewer who is not familiar
with the precise details of the implementation, but knows about image reconstruction algorithms in general, I have a feeling that something more might be said about the kinds of artefacts that might occur in these images and the ways that they have been excluded in the processing. Some imaging artefacts have already been identified in Figure 2, for example.

Despite these minor concerns, the paper makes it very clear that the improved resolution offered by MIMO and Maximum Entropy can give real advantages and insights, though these come at the cost of processing speed, so that this technique is not suited to real-time applications. Some pragmatic suggestions are presented, which might point the way toward a strategy for identifying intervals suitable for full image processing, based on more computationally efficient strategies. Additionally the discussion of potential tracking algorithms is interesting and is something that would be worth exploring in future studies.

---

## Author Comment (AC1) · 28 Nov 2018

We would like to thank the reviewer for his useful and positive comments. We will address them in detail in a final version. Here we would like to comment on a couple of them:

1. "By inspecting Figures 3-6, the improvement from SIMO to MIMO is significantly smaller than that between Capon and MaxEnt. Can the authors comment on it?"

The expected improvement from SIMO to MIMO is about 50% due to the resulting antenna configuration by using MIMO. See Fig. 1. It could be further improved by

using a longer separation between the transmitting antennas but we are limited to the currently installed antenna array. This improvement can be quantitatively verified in Table 2 for a point-like target. As described in P.7 line 1, the Capon performance is strongly affected by the increase in the number of targets. For the two events showed in this paper the illuminated area is full-filled with PMSE, so a poor performance is expected when Capon is used. However, we still expect an improvement of 50% from SIMO-Capon to MIMO-Capon. In the case of MaxEnt, the algorithm is limited by the amount of information contained in the image (Entropy), i. e., changes in the image. MaxEnt chooses the most uniform solution (image) of all possibles.

2. "The small features drifting with winds in event 1 seem like so-called ripples from gravity wave breaking. Ripples often show up in airglow and PMCs."

We agree that the description of ripples fit with our observations (Event 1). We will modify our text accordingly, including the suggested reference.

---

## Author Comment (AC2) · 30 Nov 2018

We thank the referee for his valuable and positive comments and suggestions. We think they will help to improve the quality and understanding of this paper. Here we would like to comment on some of the concerns.

1) The results clearly show intensity variations in the PMSE layers corresponding to wavelike activity, which are plausibly linked to generation by the Kelvin-Helmholtz Instability and display wind-related dynamics. This dynamics is, however, somewhat puzzling, because in one example the waves seem to drift with the background wind, while in a second case they do not.

[Figure]

R: As a first approximation, we attribute the first event to a KHI event (drifting with the wind) and the second event to a propagating gravity wave event (not drifting with the wind). Further investigation of this data is needed to explain the physical mechanisms behind these two events. This explanation is not included in this paper, however, we are planning to explore this data with more detail in the near future.

2) I found the discussion about the relationship between the phase front orientation, the drifting of the wave field and the strength and direction of the background wind somewhat confusing, because I was unsure exactly how to interpret Figure 7. The text seems to indicate that the "arrow slope" indicates the magnitude of the wind velocity, when this is normally the arrow length. Hence I am unsure how to interpret the arrow length and direction in terms of vector velocity.

R: We will modify the text accordingly to make this statement clearer. The main reason why we didn't use the standard convention ("arrow length") is that the plot axis is "Time" vs "Distance". Usually, the arrow length convention is used in plots "Distance" vs "Distance", where the arrow length indicates how much a point displaced in each direction for a certain time. In Figure 7,b the Y-axis is distance (East-West) and the X-axis is time so the velocity is equal to delta(y) divided by delta(x), the arrow slope. For example, analyzing Fig 7(b) the arrow indicates that a point displaced approx. -9km in the Y direction and 5min in the X direction, it means the velocity is equal to -9km/5min = -30m/s. We will add a table with the wind values to avoid miss-interpretation.

3) For example in event 1, the wind is apparently northward, but in Figure 7(b) the zonal wind vectors also appear substantial (at least the arrows are long in Fig 7b, which shows the zonal component). Also, in Figures 7(b) and 7(c), there appears to be wave front structure in both the meridional and zonal directions, whereas one might expect a KHI wave field driven by a meridional wind to have zonally-oriented phase fronts. I think this figure needs a clearer explanation to make it more intuitive to the reader.

[Figure]

R: We will explain better these results in the revised version. In the KHI event for example, indeed the displacement of the structures and the wind coincide, in the zonal component, since there is a wave structure (a finite wavelength). In the case of the meridional component, the structure elongated almost across the field of view, without noticeable smaller scales. Therefore, the expected meridional drifting of the structures is not clearly observed within our field of view.

4) Nonetheless the results are clearly very interesting and seem to offer significant potential for a more physics-based study. The MIMO technique combined with Maximum Entropy imaging clearly shows smaller structures than the SIMO techniques, or even MIMO plus Capon (such small structure are notable in Figures 3d, 8c, and 9c), the authors should perhaps say something about their persistence and statistical significance.

R: We are showing a conservative version of our results by using a relatively large SNR threshold, therefore increasing the statistical significance of our results. The persistence of our results can be clearly observed in the animations attached to the paper, where structures are persistent in time and space modulated by the governing background dynamics. As in any ill-posed inverse problem, the algorithm used works under certain conditions or assumptions. In the case of Capon, the performance will get worse as the number of targets increase. It is already known that in a full-filled scenario the errors and artifacts using Capon are high. On the other hand, when MaxEnt is used the errors and artifacts are low if the image is uniform, even if it is full-filled. As long as the image is more uneven the errors and artifacts will increase. The only way to avoid this problem is having more measurements than unknowns. In the future, we are planning to use Compressed Sensing (CS). The idea of CS is to find a domain ("sparse domain") where the number of unknowns (non-zero values) is less than the number of measurements. The key point here is to find the most suitable Sparse Domain or Dictionary for our data. Previous works have used a Wavelet domain as a dictionary having similar results than MaxEnt. We want to find a Wavelet-like domain

optimized for our atmospheric images to improve the results. We will add a comment on this in the discussion.

5) The text implies that some of them may be meteor echoes, but this point is not discussed in detail.

R: Meteor echoes could indeed be observed in the PMSE region, but the great majority of them occur outside this window. When a meteor echo occurs in the PMSE altitude region, which is seldom, they will be short-lived (less than a few 100 milliseconds). Their effect can be easily removed. In previous studies without imaging they were removed just from their time occurrence, now with imaging, we can exclude them from both time and angular occurrence.

6) As a reviewer who is not familiar with the precise details of the implementation, but knows about image reconstruction algorithms in general, I have a feeling that something more might be said about the kinds of artifacts that might occur in these images and the ways that they have been excluded in the processing. Some imaging artifacts have already been identified in Figure 2, for example.

R: Thanks for the good suggestion. We will add some text about the technique and the possible artifacts in the final version, as we mentioned above, we are considering a conservative approach by using a relatively large SNR threshold. As a general comment, we can see two main issues in the imaging problem (1) Point Spread Function-(PSF) and (2) image smearing. In the case of PSF, ideally one would like to have a delta function, but in practice, a PSF will have sidelobes, that could create angular artifacts. With MIMO we are improving the PSF by reducing the sidelobes and making the mainlobe narrower. Image smearing has not been included in this work and it will be analyzed in a future work, but it is basically due to the drifting of the structures as they are being imaged.

---

## Author Response (AR1)

December 10, 2018

Dr. William Ward

Editor-in-Chief
Atmospheric Measurements Techniques

Reference: Response to the referees

Title: "Enhancing the spatio-temporal features of polar mesosphere summer echoes using coherent MIMO and radar imaging at MAARSY"

Dear Editor,

We thank the reviewers for their time. We believe that their valuable comments and feedback have helped us to improve the quality and understanding of the work. Below please find a point-by-point response to the reviewer's comments. For clarity, comments are given in italics and our responses in bold and plain text.

Sincerely yours,

Juan M. Urco C.
Leibniz Institute of Atmospheric Physics at the Rostock University
urco@iap-kborn.de

**Response to Reviewer 1:**

*This paper introduces the new technique of MIMO and two radar imaging methods to improve the spatial and temporal resolution of PMSE imaging. Two PMSE events are presented. One shows small features drifting with winds, and another displays gravity waves. The paper is well written. I could recommend its publication after minor revisions.*

*1. By inspecting Figures 3-6, the improvement from SIMO to MIMO is significantly smaller than that between Capon and MaxEnt. Can the authors comment on it?*

**R: The difference is attributed mainly to the volume-filling scenario for both events which degrades the Capon performance, and the use of the statistical uncertainties by MaxEnt. In cases of reduced number of targets with relative high SNR, i.e., number of unknowns comparable with number of measurements, MIMO outperforms SIMO using both Capon and MaxEnt. We rephrased the paragraph in page 7, lines 1-15, to clarify this point.**

*2. The small features drifting with winds in event 1 seem like so called ripples from gravity wave breaking. Ripples often show up in airglow and PMCs. Hecht (2004), Instability layers and airglow imaging, Review of Geophysics.*

**R: We modified the paragraph in page 8, lines 13-32, to describe these small features and we added the suggested reference.**

*3. page 1, line 10 "goodness" –> "advantage"*

**R: Done**

*4. page 3, line 16, "e.g., see Latteck..."*

**R: Done**

*5. page 10, line 9, "e.g., M. 2013"?*

**R: Fixed**

*6. page 10, line 14, "475 m instead of 163 m".*

**R: Skipped, in this case we are talking about the number of independent samples, i.e., 475 samples instead of 163 samples.**

*7. The fond sizes in Figure 4 and 6 need to be enlarged.*

**R: Done**

**Response to Reviewer 2:**

*This is an interesting paper which shows convincingly how the effective resolution of radar targets with large backscatter cross-section, such as PMSE layers, can be significantly improved by modularising the radar aperture and using MIMO (multi-input, multi-output techniques) combined with high-resolution image processing strategies, such as Capon and Maximum Entropy. The resulting improvements in image resolution are quite striking, even for a relatively small modular radar aperture such as MAARSY, and clearly open up the possibility of understanding the structure and dynamics of PMSE layers at higher temporal and spatial resolution, with the potential to add to the current understanding of the physics of these layers. The techniques themselves are well described and in the case of the image processing strategies, have been extensively documented in previous literature. The implementation is seemingly based on earlier experiments at Jicamarca, which have attracted quite a lot of interest in the community.*

*1. The results clearly show intensity variations in the PMSE layers corresponding to wavelike activity, which are plausibly linked to generation by the Kelvin-Helmholtz Instability and display wind-related dynamics. This dynamics is, however, somewhat puzzling, because in one example the waves seem to drift with the background wind, while in a second case they do not.*

**R: We modified the paragraph in page 8, lines 13-32, to describe these small features and we added a reference which describes similar structures observed by airglow imagers. In addition, not all the ripples are associated to KHIs. In the case of event 2, given that the ripples propagate against the horizontal wind, we are speculating that they are due to propagating gravity waves.**

*2. I found the discussion about the relationship between the phase front orientation, the drifting of the wave field and the strength and direction of the background wind somewhat confusing, because I was unsure exactly how to interpret Figure 7. The text seems to indicate that the "arrow slope" indicates the magnitude of the wind velocity, when this is normally the arrow length. Hence I am unsure how to interpret the arrow length and direction in terms of vector velocity.*

**R: The main reason why we didn't use the standard convention ("arrow length") is that the plot axis is "time" vs "distance". Usually, the arrow length convention is used in plots "distance" vs "distance". We rephrased the paragraph in page 8, lines 1-7, to make this statement clear, i.e. the slope represents velocity (space vs time).**

*3. For example in event 1, the wind is apparently northward, but in Figure 7(b) the zonal wind vectors also appear substantial (at least the arrows are long in Fig 7b, which shows the zonal component). Also, in Figures 7(b) and 7(c), there appears to be wave front structure in both the meridional and zonal directions, whereas one might expect a KHI wave field driven by a meridional wind to have zonally-oriented phase fronts. I think this figure needs a clearer explanation to make it more intuitive to the reader. Nonetheless the results are clearly very interesting and seem to offer significant potential for a more physics-based study.*

**R: The text in page 8, lines 13-32, was modified accordingly to clarify this point.**

*4. The MIMO technique combined with Maximum Entropy imaging clearly shows smaller structures than the SIMO techniques, or even MIMO plus Capon (such small structure are notable in Figures 3d, 8c and 9c), the authors should perhaps say something about their persistence and statistical significance.*

**R: From single frame images is difficult to determine if a structure is real or an artifact of the technique, particularlly when there are strong signals close by. However, by analyzing the "keograms" and the movies (supplement) material, one can see that the majority of the features persist in both time and space, given us confidence on their statistical persistence. Given that we are dealing with an underdetermined problem some artifacts are expected. In order to reduce the artifacts and increase the statistical significance of our results we are using a conservative SNR threshold to filter the data. A detailed explanation was added in the discussion, page 10, lines 27-32.**

*5. The text implies that some of them may be meteor echoes, but this point is not discussed in detail.*

**R: Meteor echoes are observed by the MAARSY radar at PMSE altitudes occassionaly. Given that they occur in a localized angle and during a short time period, either their contributions are average out, or they appear as single points that can be easily identified and removed. An example can be observed in Fig. 3d. An additional paragraph was added in page 10, lines 3-6.**

*6. As a reviewer who is not familiar with the precise details of the implementation, but knows about image reconstruction algorithms in general, I have a feeling that something more might be said about the kinds of artefacts that might occur in these images and the ways that they have been excluded in the processing. Some imaging artefacts have already been identified in Figure 2, for example.*

**R: A new paragraph was added in page 10, lines 27-32, to explain the expected artifacts and the way how they are discarded.**

*7. Despite these minor concerns, the paper makes it very clear that the improved resolution offered by MIMO and Maximum Entropy can give real advantages and insights, though these come at the cost of processing speed, so that this technique is not suited to real-time applications. Some pragmatic suggestions are presented, which might point the way toward a strategy for identifying intervals suitable for full image processing, based on more computationally efficient strategies. Additionally the discussion of potential tracking algorithms is interesting and is something that would be worth exploring in future studies.*

**R: We are already working on this topic and we hope to have some results in the near future.**

[revised manuscript text omitted]

where $\epsilon$ is the noise amplitude associated with the visibility measurements. In this work, we have also considered the improvements of Hysell and Chau (2006). Specifically, we have taken into account the transmitting beam pattern and the statistical uncertainties of all the visibility pairs.

**4 Results**

Figure 2 shows the resulting 24-bit Range-Time-Doppler-Intensity (RTDI) image of the vertical beam for 32 hours of continuous operation on July 16 and 17, 2017. This plot was obtained after applying MaxEnt to the data and selecting the values that belong to the zenith angle. The signal intensity is represented as lightness, Doppler information as hue, and spectral width as saturation. As shown later, the resulting HPBW for this experiment is <1° indicating that the Doppler information must be mainly due to the vertical motion. The RTDI plot indicates that the vertical motion is slow (green color) as expected. Nevertheless, there are two regions at 23:30LT and 06:30LT around 89km where the Doppler velocity present unrealistic values. Indeed, PMSE were too strong at that time so that even the antenna sidelobes can be seen. Unfortunately, the imaging algorithm cannot assign the correct angle of arrival to these unusually strong echoes due to the angular ambiguity associated to our antenna array. The angular ambiguity is defined by the minimum separation between two antennas. The smaller the separation the larger the angle without ambiguity (e.g., Woodman, 1997). A manual angular correction can be applied knowing the Doppler but it is a hard task in the presence of many targets. A smaller baseline is recommended in future experiments for these special cases.

**4.1 SIMO vs MIMO Results**

Since the estimated brightness is expressed in polar coordinates, $(\theta_x, \theta_y, R)$, a cubic spline interpolation was applied to convert them to Cartesian coordinates, $B(\theta_x, \theta_y, R)$ to $B(x, y, z)$, with the radar being located at the center (x=0,y=0,z=0). Below we

show the results of two selected events (Event 1 and Event 2) after performing such interpolation. For both events, we show x vs y cuts for a given z, as well as x vs z cuts for a given y. Where x, y, and z represent the East-West (EW) direction, North-South direction (NS) and Altitude, respectively.

Examples of EW-NS and EW-Altitude 2D images for Event 1 obtained by applying Capon and MaxEnt to two different antenna configurations, SIMO and MIMO, are shown in Figs. 3 and 4, respectively. Four different results are shown (a) SIMO-Capon, (b) SIMO-MaxEnt, (c) MIMO-Capon, and (d) MIMO-MaxEnt. Having a quick look at the results is clearly observable that: (1) MaxEnt outperforms Capon when the same antenna configuration is used, either SIMO or MIMO. This was already pointed out by previous works (e.g. Yu et al., 2000; Harding and Milla, 2013). (2) As expected, MIMO shows a cleaner and more defined image compared to SIMO, when either Capon or MaxEnt is  employed. (3) The improvement of using MIMO instead of SIMO, is much better in MaxEnt than Capon. The improvement of MIMO-Capon  with respect to SIMO-Capon  is about 50% due to the larger virtual antenna array. Whereas, the improvement of MIMO-MaxEnt with respect to SIMO-MaxEnt is much better than 50%. This difference lies in the fact that Capon tries to reduce the  sidelobes adaptively steering them to echo-free zones. Unfortunately,  for the two events shown most of the illuminated area is filled with PMSE scattering and thus the performance of Capon is  expected to be comparable to the conventional beam forming (Inverse Fourier Transform). In the case of MaxEnt, the improvement is mainly due to the larger virtual antenna array and the use of statistical uncertainties as described by Hysell and Chau (2006). Unlike Capon, our MaxEnt implementation takes advantage of the redundant visibility pairs giving more weight to pairs with less uncertainty, i.e. more redundancy. Fig. 1b and 1e show the visibility pairs and their redundancy for SIMO and MIMO, respectively.

Coming back to our comparison SIMO vs MIMO, with MIMO-MaxEnt small wave-like structures of 2 km wavelength can be clearly observed, which are invisible in SIMO implementations or MIMO-Capon. For example, observe the two wavefronts at x=10 km in Fig. 3d, right beside the larger  meridionally-oriented wavefronts of 7 km wavelength.  This indicates that wave-like structures of different wavelengths coexist within PMSE as previously seen in NLC (e.g., Baumgarten and Fritts, 2014). In addition, Fig. 4d shows that the ascending  structures (red color) have higher SNR than the descending  structures (blue color).

We show similar 2D cuts for Event 2, in Figs. 5 and 6 for z=82.7 km and y=-6 km, respectively. In this case, the observed wavelength is 12 km. Unlike the first event, the SNR is similar for targets with negative and positive Doppler. Figure 6d shows two very interesting points: (a) a very well defined wave-like structure between 82-84 km and (b) a quasi-uniform structure between 84-86 km which apparently has been modulated by the first wave. In this case, the wave-like structure is easily discernible even with SIMO-Capon, given that the wavelengths are larger than in Event 1 (see Fig. 5a).

**4.2 MIMO results**

Having shown the better qualitative performance of MIMO-MaxEnt with respect to the other three implementations for the two selected events above, next we present extended results using just MIMO-MaxEnt.

Figure 7 shows the evolution in time of two selected events, i.e., Event 1 (left), and Event 2 (right). Figures 7a and 7d show
5  the  time evolution vs Altitude for selected EW and NS coordinates. In these plots, we can appreciate how variable PMSE structures are, showing different altitudinal extensions. Note that the effective horizontal area is less than $1\ km^2$ in both cases.

The second and third row of Fig. 7  show the time evolution vs EW direction and the time evolution vs NS direction, EW and NS keograms respectively. We have included the zonal ($u_0$) and meridional
10  ($v_0$) wind velocity estimated from combining a couple of specular meteor radars (SMRs) (pink arrow) and from MAARSY based on PMSE Doppler velocities (yellow arrow). The wind values are shown in Table 3. Since this is a plot "time" vs "distance", the zonal and meridional wind are represented by arrows where their slope indicate the wind magnitude, i.e, how long a target is displaced in the Y-axis for a certain time in the X-axis. The SMR winds were obtained from combining SMR detections from Andenes and Tromso in northern Norway (e.g., Chau
15  et al., 2017, for details). In order to estimate the winds from PMSE we used the following formula:

$$v_{rad}(\theta_x, \theta_y, \theta_z) = u \cdot \theta_x + v \cdot \theta_y + w \cdot \theta_z \tag{7}$$

where $v_{rad}$ is the radial wind, $(\theta_x, \theta_y, \theta_z)$ are the direction cosines and $(u,v,w)$ are the zonal, meridional and vertical wind direction, respectively. Assuming a constant $u,v$, and $w$ for a given altitude bin and time bin, and taking all the measurements with SNR higher than -5dB we  invert Eq. 7 and get $u_0$, $v_0$ and $w_0$, mean values of u, v and w respectively.
20   The keograms for Event 1, i.e., from 00:50 to 01:05 UTC, show that the meridionally-oriented wavefronts have a limited vertical extent centered at 85 km, Fig. 7a,  and since this wave has a finite wavelenght in the EW direction the zonal wave propagation can clearly be observed in Fig. 7b,  where the elongated meridionally-oriented wavefronts are zonally drifting
25  with the same direction and with the same speed as the wind. In the NS direction,  the meridional drifting of the wave is not clearly observed due to the elongated structure. Mesospheric wave-like features observed with airglow imagers (ripples) have been also noticed to drift with the background ~~. Looking carefully at the first front wave reaches y=6 km, the
30  second one reaches y=10 km and the last one a bit more. Unfortunately, our limited illuminated area width does not allow us to corroborate it.~~ wind (Hecht, 2003, e.g.). These ripples have been associated to gravity wave breaking and are a clear signature of atmospheric instability.

Figure 7d shows another interesting  wave-like example (Event 2). Unlike the first case, this wave does not keep its amplitude in  the vertical direction, see Fig. 7d. It grows and then disappears. Its direction of propagation in  the zonal and meridional direction is also interesting. As shown in Fig. 7e, the direction of propagation in the  zonal direction is completely opposite to the background wind. Whereas the wind is going from East to West, the wave propagates from West to East. In the NS direction, Fig. 7f, the wind is close to zero and we do not expect changes in this direction. Since its wavelength is relatively small, this structure might be classified as an instability, however, the opposite direction of propagation suggests that it could be a propagating gravity wave. Further investigation of these events is needed to understand the physical mechanisms behind them, including lidar and airglow imager observations.

PMSE has been used as a neutral wind tracer assuming that u, v and w are constant and homogeneous during the analyzed time (e.g., Balsley and Riddle, 1984; Fritts et al., 1990; Hoppe and Fritts, 1995; Stober et al., 2013). Therefore, those works assumed that scatters from PMSE are moving with the neutral wind at the same velocity and in the same direction. Unlike winds obtained from SMR, winds from PMSE are affected by local disturbances as shown in Figs. 7e and 7f. When the dynamics of local structures are not in agreement with the wind dynamics a bias could be introduced in the wind estimation (as shown in the Event 2). However, when these local disturbances are moving with the wind the estimated wind is not affected (Event 1). Note that the PMSE winds are in good agreement with the SMR winds in Event 1, but they are not for Event 2, particularly for the meridional component.

An animated sequence of the two events have been included as supplemental material, i.e., Movie S1 and S2. For both events, the sequence includes selected cuts of EW-NS, EW-Altitude, and NS-Altitude. In the Movie S1, we identify at least four examples of monochromatic waves with different wavelengths drifting with the wind in the direction North-West (at 23:57:37, 00:02:24, 00:10:57, 00:55:33 UTC). Interestingly in this case, longitudinal and transverse waves both drift with the background wind. In Movie S2, we show the complete evolution in time of Event 2. In the EW-Altitude cut, the wave structure between 82-85 km drifts against the wind, whereas a layer at 87 km between 05:20 and 05:30 UTC follows the background wind. Note the projected radial wind (from red to blue) indicating a westward wind. These events are good examples of the complicated dynamics within PMSE. Further analysis and interpretation of these high-resolution spatiotemporal structures will be done on a future work.

Figures 8 and 9 shows 3D maps of (a) the signal to noise ratio (SNR) (b) radial velocity, (c) locally enhanced SNR, and (d) residual radial velocity (i.e., $v_{res}$), for Events 1 and 2, respectively. In addition contours of locally enhanced SNR ae overplotted on both the radial velocities. The SNR and radial velocity were obtained from the first and second spectral moments (e.g., Doviak and Zrnić, 1993). The locally enhanced SNR has been obtained using a 2D Gaussian function kernel with a width of 6 pixels. The local enhancements allow us to observe weak structures within strong one. For example, wave fronts are distinguishable in Fig. 8c which were not visible in Fig. 8a. On the other hand, the residual radial velocity was estimated by removing the contributions of the estimated mean horizontal velocities in the measured radial velocities, i.e.,

$$v_{res}(\theta_x, \theta_y, \theta_z) = v_{rad}(\theta_x, \theta_y, \theta_z) - (u_0 \cdot \theta_x + v_0 \cdot \theta_y) \tag{8}$$

Assuming that the $v_{res}$ is mainly due to the vertical motion, we can clearly see in Fig. 8d how up (red) and down (blue) structures drift across the illuminated area, maybe due to KHI. Similarly, Fig. 9 shows animated images of Event 2. In this case, the horizontal wind was small and most of the radial velocity was due to the vertical motion, i.e., radial velocity and residual velocities are almost the same. As mentioned above, in this event, the waves propagate horizontally against the weak horizontal wind.

The animated versions of Figs. 8 and 9 are shown in the supplemental material Movies S3 and S4, respectively. Although the information might be redundant when compared to Movies S1 and S2, we have decided to include them to provide a more standard view of typical spectral parameters of a multi-beam radar.

Making a quantitative comparison between SIMO and MIMO for real targets is not an easy task. We need a prior knowledge of the brightness to make a good analysis. This is not the case for PMSE. Fortunately, our observations include echoes from specular meteors, see the bright echoe located at (-10.5,-12.5) in Fig. 3d. Indeed, meteor echoes can be observed in the PMSE region, but the great majority of them occur outside this window. When a meteor echoe occurs in the PMSE altitude they will be short-lived (less than a few hundred miliseconds). In previous studies meteor echoes were treated as outliers and were removed from the measurements (e.g., Hashimoto et al., 2014). For our benefit they can also be used to evaluate quantitatively the angular resolution that can be achieved with our implementations. A specular meteor echo could be considered as a point target in angle. Along with its trajectory, the trail is long (hundreds of meters to a few kilometers) but its angular response is narrow. In the transverse direction to the trail, it is very narrow and its angular response is also narrow.

In Fig. 10 we show the normalized angular scattered power distribution for a specular meteor using SIMO and MIMO in combination with Capon and MaxEnt. As expected, the range resolution does not change for SIMO or MIMO [see Fig. 10(a)]. We see a peak at 89.1km and low power at other ranges. However, when comparing Capon and MaxEnt, MaxEnt shows us a clean power distribution along the whole ranges while Capon shows us a remaining sidelobe contamination at other ranges, coming from other angles. This indicates that, even with MIMO, Capon does not suppress the sidelobes as well as MaxEnt. Figures 10b and 10c show us the angular power distribution for $\theta_x$ and $\theta_y$ respectively, where the points are the samples for a given angle and the continuous line is a fitted Gaussian function. Using the fitted function we estimated the half power beam width (HPBW) for each implementation. Table 2 summarizes the angular resolution and the improvement factor for each method compared to the theoretical angular resolution of the full array MAARSY radar. As we expected the improvement between SIMO and MIMO is about 1.5 given that we increased the antenna aperture for MIMO by $\sim 50\%$. When combining MIMO and MaxEnt, surprisingly, we got an angular resolution of $\sim 0.6°$, i.e., more than 5 times better than MAARSY's HPBW.

**5   Discussion**

We have shown qualitatively and quantitatively that radar imaging of PMSE is significantly improved by using MIMO instead of SIMO configurations, by at least $50\%$. Two different imaging methods have been applied, Capon and MaxEnt. As expected

from previous works, MaxEnt images are better than Capon images, however, MaxEnt is computationally more demanding. Similarly, we found that the quality of MIMO-Capon is comparable to SIMO-MaxEnt.

Even though MIMO allows us to improve the point spread function it is not perfect. We expect some artifacts due to the sidelobes which are -15dB weaker than the mainlobe, see Fig. 1f. When strong and weak echoes coexist in the same region

5 some artifacts might be confused as weak echoes. Although Capon and MaxEnt help to minimize the sidelobe contribution, we are being conservative by employing a relatively large SNR threshold (>5dB), i.e., discarding weak echoes which might be contaminated by strong sidelobe echoes. By doing this we are increasing the statistical significance of our results which are persistent in time and space as shown in the animations and the keograms.

[revised manuscript text omitted]

**Figure 2.** A 24-bit image of range-time Doppler intensity (RTDI) plot of PMSE using MIMO with time diversity conducted on July 16 and 17, 2017. The signal intensity is represented as lightness, Doppler information as hue, and spectral width as saturation. The legend on the left represent the SNR vs Doppler color map for a saturation of 90%. The legend on the right represents the spectral width vs Doppler for a lightness of 50%. Note that only the signal corresponding to the narrow region in the illuminated area is shown.

[Figure]

**Figure 3.** EW-NS images for z=85.8 km obtained from applying four different implementations (a) SIMO-Capon, (b) SIMO-MaxEnt, (c) MIMO-Capon, (d) MIMO-MaxEnt, at 00:56:55 UT on July 17, 2017, i.e., Event 1. Images are color coded as same as Figure 2 The yellow dashed horizontal/vertical lines represent the location of the NS/EW cuts shown in later figures for Event 1.

[Figure]

**Figure 4.** Similar to Fig. 3, but for an EW-Altitude cut at y=0 km. The yellow dashed horizontal lines represent the location of the altitude cuts shown in previous and later figures for Event 1.

[Figure]

**Figure 5.** Same as Figure 3 but at 05:56:13 UT on July, 2017, i.e., Event 2. The yellow dashed horizontal/vertical lines represent the location of the NS/EW cuts shown in later figures for Event 2.

[Figure]

**Figure 6.** Same as Figure 5 but for an EW-Altitude cut at y=-6 km. The yellow dashed horizontal lines represent the location of the altitude cuts shown in previous and later figures for Event 2.

[Figure]

**Figure 7.** 24-bit time representation images of PMSE structures as function of: altitude (RTDI) (first row), EW location (Keogram) (second row), and NS location (Keogram) (third row) for selected cuts, for both Event 1 (first column) and Event 2 (second column). In the keograms, the wind components obtained with specular meteor radars (SMRs) and MAARSY PMSE are shown with pink and yellow arrows, respectively. The white dashed horizontal lines represent the location of the altitude, EW and NS cuts shown in previous figures and current keograms for Event 1 and 2. The white dashed vertical lines represent the time of the cuts shown in previous figures.

[Figure]

**Figure 8.** Thre-dimensional contour plots at 00:55:33 UT on July 17, 2017, i.e., Event 1, for four selected altitudes : 84km, 84.6km, 85.2km, and 85.8km. For each altitude is shown: (a) SNR, (b) radial velocity, (c) Locally enhanced SNR, and (d) residual radial velocity. Contour on locally enhanced SNR are overplotted in both velocity plots.

[Figure]

**Figure 9.** Same as 8, but at 05:54:01 on July 17, 2017 for altitudes: 82km, 82.7km, 83.4km, and 84km., i.e., Event 2.

[Figure]

**Figure 10.** Normalized angular power distribution of a specular meteor echo as function of: (a) range, (b) EW angle ($\theta_x$), and NS angle ($\theta_y$). The results are shown for all four implementations, i.e., SIMO-Capon (blue), SIMO-MaxEnt (orange), MIMO-Capon (green), and MIMO-MaxEnt (red).

**Table 1.** Parameters of MAARSY MIMO experiment

| Parameter | Value |
|---|---|
| Frequency | 53.5MHz |
| Pulse repetition frequency (PRF) | 1000Hz |
| Pulse coding | Complementary 16 |
| Number of transmitters (beams) | 5 |
| Transmit diversity | Time |
| Tx interleaving | 2ms |
| Number of coherent integrations | 8 |
| Effective PRF (after integration) | 25Hz |
| Number of FFT points | 16 |
| Number of incoherent integrations | 128 |
| Equivalent integration time | 81.92s |
| Range resolution | 450m |

**Table 2.** Performance of imaging techniques

| Technique | Angular resolution | Spatial resolution at 85km | Equivalent antenna aperture | Improvement factor |
|---|---|---|---|---|
| MAARSY | 3.60° | 5.33km | 76m | 1.00 |
| SIMO - Capon | 1.27° | 1.88km | 216m | 2.83 |
| MIMO - Capon | 0.88° | 1.30km | 312m | 4.09 |
| SIMO - MaxEnt | 1.05° | 1.55km | 261m | 3.42 |
| MIMO - MaxEnt | 0.61° | 0.90km | 450m | 5.90 |

**Table 3.** Mean wind values for the two events presented

| | EVENT 1 | | EVENT 2 | |
|---|---|---|---|---|
| | PMSE | SMRs | PMSE | SMRs |
| Zonal wind - $u_0$ (m/s) | -33.4 | -29.15 | -12.20 | -17.19 |
| Meridional wind - $v_0$ (m/s) | 17.80 | 21.96 | 5.17 | -0.22 |